# Impact of brining and drying processes on the nutritive value of tambaqui fish (*Colossoma macropomum*)

Awot Teklu Mebratu [1,2]*, Yohannes Tekle Asfaw[3], Wouter Merckx[4], Wouter H. Hendriks[5], Geert P. J. Janssens[1]

**1** Department of Veterinary and Biosciences, Faculty of Veterinary Medicine, Ghent University, Merelbeke, Belgium, **2** Department of Animal Reproduction and Welfare, College of Veterinary Sciences, Mekelle University, Mekelle, Ethiopia, **3** Department of Veterinary Basic and Diagnostic Sciences, College of Veterinary Sciences, Mekelle University, Mekelle, Ethiopia, **4** TRANSfarm, Engineering and Technology Group, The Catholic University of Leuven, Lovenjoel, Belgium, **5** Animal Nutrition Group, Wageningen University, Wageningen, The Netherlands

* awotteklu.mebratu@ugent.be

**Data Availability Statement:** The data underlying the results presented in the study are all uploaded as Supporting information files with the paper.

## Abstract

Preservation of fish as diet ingredient is challenging in many tropical regions due to poor socioeconomic conditions and lack of freezing facilities. So, alternative preservation techniques could be viable to address the issue. The present study evaluated the effect of brine salting (15% *w/v*) prior to drying at different temperatures on the nutrient profiles of tambaqui fish (*Colossoma macropomum*). Whole fish samples (n = 48; 792 ± 16 g; 8 months old) were grouped into two as brine-salted and non-salted, and treated at seven different drying temperatures of 30, 35, 40, 45, 50, 55 and 60°C for a period of 23 h each. To evaluate the impact of Maillard reaction, reactive lysine was also quantified. Drying temperature had no effect on the evaluated macro- and micro-nutrients of tambaqui fish ($P > 0.05$) while brining reduced the overall protein concentration by 6% (58.8 to 55.4 g/100 g DM; $P = 0.004$). Brining significantly reduced many amino acids: taurine by 56% (7.1 to 3.1 g/kg; $P < 0.001$), methionine 17% (14.7 to 12.1 g/kg; $P < 0.001$), cysteine 11% (5.1 to 4.4 g/kg, $P = 0.016$), and reactive lysine 11% (52.0 to 46.4 g/kg; $P = 0.004$). However, alanine, arginine, and serine were not affected by brining ($P > 0.05$). Brining also reduced the concentrations of Se by 14% (149 to 128 µg/kg DM; $P = 0.020$), iodine 38% (604 to 373 µg/kg DM; $P = 0.020$), K 42% (9.71 to 5.61 g/kg DM; $P < 0.001$) and Mg 18% (1.32 to 1.10 g/kg DM; $P = < 0.001$) versus an anticipated vast increase in Na by 744% (2.70 to 22.90 g/kg DM; $P < 0.001$) and ash 28% (12.4 to 16.0 g/100g DM; $P < 0.001$) concentration. Neither brining nor drying temperature induced changes in % lysine reactivity and fat content of tambaqui fish ($P > 0.05$). Agreeably, results of multivariate analysis showed a negative association between brining, Na, and ash on one side of the component and most other nutrients on the other component. In conclusion, drying without brining may better preserve the nutritive value of tambaqui fish. However, as a practical remark to the industry sector, it is recommended that the final product may further evaluated for any pathogen of economic or public health importance.

**Funding:** The project was financially supported by the VLIR-UOS TEAM project under the title 'Win-win protein: reconciling fish protein generation and water quality management' [grant number ET2018TEA462A101] through Geert P.J Janssens and Luc De Meester; https://www.vliruos.be/en/home/1. The funders had no role in study design, data collection and analysis, decision to publish, or preparation of the manuscript. ATM received support in form of salary while doing the research activities.

**Competing interests:** The authors have declared that no competing interests exist.

# Introduction

Fish meat is a high-value, perishable protein-rich food requiring appropriate preservation to maintain its organoleptic properties after harvest [1, 2]. Nutritionally, fish meat is a prominent source of high quality essential amino acids [3, 4], omega-3 fatty acids such as eicosapentae-noic acid (EPA) and docosahexaenoic acid (DHA), and trace elements such as iron, selenium, iodine that are beneficial for human health [5]. Hence, good knowledge on the nutrient pro-files of fish meat enable a balanced and healthy diet for the human consumer and endorses consumer preferences on dietary requirements [5, 6].

Tambaqui fish (*Colossoma macropomum*), also known as black pacu, is the main native fish farmed in South and Central American countries [7]. It is mainly found in Brazil, Venezuela, Colombia, Peru, and Bolivia. Tambaqui alone accounted for a total production of 142,135 tons in 2016 with Brazil accounting for about 96.4% of the total production [8]. Its life cycle is mainly associated with floody plains of white-water rivers in the amazon basins that are char-acterized with a temperature range of 25–34°C, low to neutral pH (6–7), and rich in suspended solids [9, 10]. Tambaqui is an omnivorous fish that feeds on a variety of natural feeds such as fruits and seeds, and zooplanktons depending upon seasons [8, 11]. It is considered to be a nocturnal feeder fish [7]. Nutritionally, the tambaqui is also rich in essential amino acids, EPA and DHA, selenium, and vitamins D and B [9]. Furthermore, tambaqui requires a fairly low protein content in its feed, most of which (75–85%) can be provided as plant protein [7, 12]. This makes it possible to breed tambaqui with the aid of low-cost feed and feedstuffs, such as forest fruits or seeds of leguminous plants [7, 13].

Food preservation is a fundamental requirement in maintaining quality food, a vital issue in human health, and contributes to profitability [14, 15]. Fish meat, for instance, contains of nutrients that can easily get exposed to the ubiquitous environmental microbials (bacteria, moulds,...) and chemicals causing degradation. The moisture content of fish meat predisposes for degradation and affects the nutritive value and shelf life of the product [16]. Fish meat often has a high moisture content averaging 70–84% where bacteria can grow rapidly and cause spoilage [4]. The suppression or killing of such microorganisms can be achieved by either reducing the temperature (e.g. chilling, freezing) or reducing the water content (e.g. dry-ing, salting) [15]. Drying of fish meat products can be achieved by traditionally sun-drying or open air drying and using air driers, or microwaves [5, 17]. Salting, on the other hand, involves different processes techniques such as brining (the immersion of fish meat into salt solution), kench curing (dry salting of fish fillets), pickling (longer storage of the fish with brine), and the possible injection of brine into the fish muscle all ending up in extended product shelf life [5, 18–21].

Brine salting is one of the oldest traditional and economically viable preservation methods of edible animal proteins such as fish meat [22–24]. It involves the use of salt (NaCl) that, due to its hygroscopic property, withdraws water from tissues via osmosis and, thereby, reduces water activity to slow bacterial growth [22]. In addition, salt is also reported to improve prod-uct flavor, extract proteins, aid in emulsion formation, and reduce proteolytic activities of enzymes [25–27]. The quantity and concentration of salt and treatment time are determining factors in the process that could affect the nutritional quality of the fish [28–30]. A 15% brine concentration for a period of 48 h has been reported to yield optimum protein solubility/avail-ability in a comparative study carried out in cod [24]. Likewise, better mass transfer kinetics of salt to tissues of cod and salmon were observed at 15% (*w/w*) brine concentration [23]. In fact, 10–15% (*w/w*) is the salt concentration recommended for salting whole fish under the food safety and standards regulation [31]. Higher concentrations of up to 25% (*w/w*) denature mus-cle proteins by causing cell membrane disruption, decreased enzymatic activity, or interaction

with myofibrils, and reducing its water-holding capacity [23, 24, 32]. In general, higher yields of tissue proteins are obtained at lower salt concentrations [24, 26, 33].

Drying involves the exposure of a product to specified temperature and time period to facilitate loss of tissue water to a lower limit of water activity ($a_w < 0.90$) where microbial growth is inhibited [34]. Drying temperature, relative humidity and air velocity are determining the rate of dehydration [2]. In rural areas, temperature-controlled ovens are usually not available, and open-air drying is the only option. Especially high environmental temperatures during drying may affect the nutritive value of fish, for instance because heat may damage amino acids or may increase the reaction of amino acids with lipids and/or carbohydrates to form biologically unavailable Maillard products [35]. To determine the risk of drying at uncontrolled temperature for the nutritive value of fish, it is warranted to study the effect of a wide enough temperature range for fish drying.

Depending on the type of drying oven and species and size of the fish involved, a temperature range of 65–135°C and drying time of 2–20 h have been routinely used for drying of fish meat [2, 26]. However, due to possible conjoint loss of some non-moisture compounds (e.g. volatile fatty acids and vitamins), lower temperatures with extended drying time are often recommended [36]. For example, drying of salted shark at 35°C reported to yield improved protein solubility, tissue rehydration and water retention capacity and total protein estimates compared to 45, 55 and 60°C [37]. Pacheco [38] also stated that the maximum drying temperature for the majority of the edible fish meat should not exceed 70°C. In fact, drying of fish products at higher temperatures, of more than 60°C, reported to cause fat oxidation and caramelization of sugars [37].

Despite the effect of brining and drying temperatures on the nutrient profiles of different fish species has been investigated, we hypothesized that species differences in nutrient profiles could respond differently to drying and brining. We here evaluated tambaqui fish. Our study also investigated reactive lysine as a marker for Maillard product formation: lysine is a limiting amino acid in most diets and needs to be bioavailable to be used in metabolism. Despite its high concentration in tissue deposition, some reducing sugars, fats and their oxidation products and vitamins can bind the ε-amino group of lysine and generate Maillard reaction products during fish processing [35]. Hence, measuring the content of "reactive lysine" as a percentage of total lysine in tambaqui fish can indicate state of protein damage during processing. The present study therefore evaluated the effect of brining and a range of drying temperatures on macronutrients and micronutrients, and mineral composition as well as on the extent of Maillard reaction that may reduce lysine availability.

## Materials and methods

### Experimental animal ethics and welfare

All experimental and animal care procedures were realized based upon the current written knowledge of aquaculture technologies and tambaquí fish farming. Fingerlings of *Colossoma macropomum* were initially brought from Aqua-Koi NV farm (http://www.aqua-koi.be/documents/home.xml?lang=nl) to TRANSfarm, Belgium. Fingerlings were then raised in a recirculating aquaculture system (RAS) with a weekly basis 30% water renewal (20 L/day) for about 7 months period before being subject to the current study. During their stay, fish were kept in larger aqua-tanks (100 cm × 120 cm × 40 cm) at a density of 1.2 fish/m$^3$ sufficient enough for movements and behavioral expression. Tanks were fixed with individual bio-filters, well-aerated with a constant temperature supply, 18:6 h light—dark cycle, and supplying pipes were regularly inspected and cleaned with tap water and water quality parameters were checked twice a week and water renewed accordingly [39].

For our experimental study, after a 12-h fasting, a total of 48 tambaqui fish were randomly selected and kept in a separate tank and were euthanized in a two-step procedure [40]. Six fish were randomly taken at a time and placed in a tank with a volume of 240 L water into which 50 mL of phenoxyethanol-solution (Anest Fish©, Germany) was added. After five minutes, 30 L of water was elevated out of the tanks and put into a large bucket. Another 6 mL of Anest Fish© was added to this bucket. Each group of six unconscious fish was scooped out of the tank and brought to the slaughter room in the bucket with an anesthetic agent (1 mL:5 L, Anest Fish©, Germany). Thereafter, tambaqui were killed with a firm blow to the head, followed by fracturing the brainstem with a knife and then transported in Ice buckets for the experiment [40]. The study was conducted in accordance with the Declaration of Helsinki and directive 2010/63/EU of the European parliament and of the council on the protection of animals used for scientific purposes. The wellbeing of the cultured tambaqui fish before being subjected to the experiment was assessed and approved by the institutional review board of TRANSfarm, KU Leuven, under laboratory accreditation of KU/LA-1210616.

## Study materials

A total of 48 tambaqui fish with a mean weight of 792 (± 16) g and 8 months old were collected from TRANSfarm, Lovenjoel, Belgium, and transported to the animal nutrition laboratory of the Department of Veterinary and Biosciences, Ghent University in styrofoam boxes with ice packs (Luxaplast, NORDIC®). Upon arrival, to facilitate drying whole fish were dissected dorso-ventrally into two equal slices using a stainless post mortem saw (SC-PM/110, 254MM) and scalpel-blades, thoroughly washed using distilled water in pipes and then individually stored in plastic bags at -20˚C for 48 h. All dissected fish slices had an average uniform dimension of 21.2 ± 2.8 cm long and 12.6 ± 2.0 cm wide and average weight of 389 ± 11.5 g. Industrial table salt (99.9%, BDH-PROLABO®, EC) was used for preparing the 15% (*w/v*) brine solution in the present study [24].

## Experimental design and set-up

Uniformly sliced whole fish samples were grouped into two as brine-salted and non-salted, and treated at seven different drying temperatures of 30, 35, 40, 45, 50, 55 and 60˚C for a period of 23 h each [23, 24, 31, 37]. Each temperature treatment was considered as a sampling point and tested in triplicate. Similarly, as a reference point, fresh whole-fish samples were initially evaluated in triplicate for their nutritional profiles.

## Brine salting and drying processes

Brine-salted fish samples were immersed in brine solution (15% *w/v*) at a ratio of 3:1 of fish to salt solution [22, 27]. For each sampling unit, samples were totally immersed and brined together in a closable plastic bucket (20 L) for 48 h at 4˚C. The brine solution was pre-cooled to 4˚C prior to the immersion of samples. Afterwards, brined samples were drained from salt drips for 3 min using a plastic sieve and randomly distributed into individual aluminium trays (500 mL; 142 mm × 116 mm × 30.4 mm; Boni™, Halle, Belgium), at one fish per tray, into which small beads (13 mm diameter, 3.5 g, JVLAB®) were added (10 beads/tray) at the bottom to avoid crusting of samples and optimize moisture loss. Likewise, the non-salted group of fish samples were randomly distributed into identical aluminium trays containing similar beads. Each tray was placed in an electric oven (EP66, BEKSO-SA®, Belgium) and randomly set and dried with the above-mentioned temperature points. The laboratory humidity was in a range of 50–60% during the experimental period. Drying temperature between sampling units were controlled using an oven-fixed thermometer and a consistent oven temperature (± 1˚C) was

maintained throughout the drying time period. A constant air ventilation was supplied through an oven-fixed ventilator at a steady air velocity of 1.0 m/s. Drying time started after oven closure and the oven returned to its set temperature (~ 5 min). After 23 h, samples were collected and allowed to cool in a desiccator for 5 min to room temperature before weighing. Samples were then kept at -21˚C overnight and finally freeze-dried (CoolSafe™, SCANVAC, Denmark) for a period of 16 h each to a stable weight, and uniformly ground (2.0 mm sieve). Subsequently, the samples were homogenized using a grinder machine (De'Longhi™, KG48) at 12× speed and then individually kept in closable plastic containers at 4˚C until analyzed [22].

## Nutrient evaluations

**Proximate analysis.** So as to make legitimate comparisons between the processed samples, the quantitative analysis of the values of macromolecules is a significant nutritional marker and for which case the proximate analysis was used to determine the crude protein (CP), ether extract and ash contents of fish samples. The dry matter (DM) content of freeze-dried samples (1 g) was determined through oven-drying at 105˚C for 24 h to constant weight [40]. Ether-extract was determined using the acid hydrolysis method [41], CP using the Kjeldahl method involving a nitrogen analyzer (LECO FP528 Instruments, St. Joseph, Michigan, USA) and N × 6.25 conversion factor [42], and ash content using an automated muffle furnace (Nabertherm™, UK) as per the procedures described by AOAC Method 938.08 [43]. A gram of fish sample was used for the analysis of each of the macronutrients and all samples were evaluated in duplicate at the animal nutrition laboratory of the Department of Veterinary and Biosciences, Ghent University, Belgium.

**Mineral analysis.** The total ash content was determined following the AOAC Method 938.08 [43] and mineral (calcium, sodium, potassium, copper, iron, magnesium, manganese, phosphorus and zinc) concentrations of fish were evaluated after dry ashing mineralization according to ISO (ICP-AES: ISO 11885) method [44] as per the procedures described by Gorsuch [45]. In brief, a gram of sample was weighed, placed into a porcelain crucible and oven-dried at 105˚C for 3 h. Sample was then burnt on a hot plate and incinerated in a muffle furnace at 450˚C for 16 h. Obtained ash was then moistened with a small amount of deionized water first and dissolved in 10 mL of 1:1 (*v/v*) solution of hydrochloric acid (HCl) and deionized water and evaporated to dry. Finally, the mix was re-dissolved in 10 mL of 1:9 (*v/v*) solution of HCl and deionized water, transferred into a 50 mL flask, and diluted to volume with deionized water. The measurement of individual mineral concentration was done using induction-coupled plasma optical emission spectrometry (ICP-OES) method as per the manufacturer's procedures (Iris Intrepid II XSP, Thermo Fisher Scientific, Aalst, Belgium). So as to obtain an optimum concentration range for the atomic absorption spectrometric method, the concentrations of certain minerals were determined after fivefold (Mg and Ca), 10-fold (Na and Zn), and 50-fold (K) dilutions. Likewise, the concentration of selenium was determined by ISO method (ICP-MS: ISO 11885) and iodine with tetramethylammonium hydroxide extraction and iodine quantification (ICP-MS: ISO 11885) at the Laboratory of Chemical analysis, Ghent University, Belgium. As quality control protocols, visual inspection of pump tubing at regular intervals (every 30 min) and replacing between samples were often made to increase the precision measurements. Similarly, regular inspection and cleaning of the injector tube, spray chamber and nebulizer, and cleaning of the widows are done to avoid contamination and reduction of light intensity. Blank solutions with similar matrix were used to improve the measurement accuracy and the detection limit set was 1–100 ppb.

**Amino acid and OMIU-reactive lysine analyses.** The total amino acid profiling was done by acid hydrolysis (2 g) at 110˚C for 23 h and ion-exchange chromatography with post-column

derivatization with ninhydrin method (ISO 13903). Sulphur-containing amino acids were measured as cysteic acid and methionine sulfone after oxidation with performic acid; tryptophan was measured by alkaline hydrolysis at 110˚C for 20 h and ion-exchange chromatography with fluorescence detection method (ISO 13904) according to procedures described by Hulshof [46].

For analysis of reactive lysine, about 2 g of fish sample were processed from each of the seven sampling points for defatting by extraction with light petroleum ether without acid hydrolysis (ISO 6492), finely ground and homogenized (2.0 mm sieve size) using a mixer mill (Retsch MM200, BV) and used for the analysis of reactive lysine via O-methylisourea (OMIU) method according to Hulshof [46]. Briefly, 1 mL of 0.6 M solution of OMIU was prepared as described by Moughan and Rutherfurd [47] that converts lysine with free e-amino group to homoarginine. Four gram of barium hydroxide octahydrate (Sigma-Aldrich, Zwijndrecht, the Netherlands) was added to 16 mL of boiling (10 min) distilled deionized water to remove $CO_2$, after which 2 g of OMIU sulphate salt was added. The solution was then cooled for 30 min at room temperature, centrifuged (6,400 × g for 10 min at 20˚C) and the supernatant was retained and the precipitate was washed with about 2 mL of same water and further centrifuged (6,400 × g for 10 min at 20˚C). both supernatants were combined and pH adjusted to 11.5 (for guanidation) by adding 6 M HCl solution. The homoarginine content was analysed in duplicate and the OMIU-reactive lysine was calculated from the homoarginine content. All amino acid profile, including reactive lysine, were determined at Wageningen University & Research, Animal nutrition laboratory, The Netherlands.

## Statistical analysis

Data obtained in the current study were analyzed using SPSS statistical software version 27 (IBM Corporation, Armonk, NY, USA). All data were checked for normality distributions using histogram and QQ-plots. A univariate one-way ANOVA was used with brining as fixed factor and temperature as covariate to test the hypothesis about differences between two or more mean nutrient values. Data were presented as means and standard error of the means (SEM). Statistical significance was set at $P < 0.05$ for any differences observed between samples. Furthermore, principal component analysis (PCA) was computed using on the collective factors obtained to appraise the relationships of nutrient profiles based on their responses towards brining and drying methods, with values deemed relevant below -0.5 and above 0.5 of the components' values.

## Results

Tambaqui fish samples were grouped into two as brine-salted and non-salted, and treated at seven different drying temperatures for 23 h each. Each temperature treatment was considered as sampling point and tested in triplicate (n = 3) and mean values obtained were statistically computed for all nutrient profiles evaluated.

### Proximate composition

Drying temperature had no significant effect on the measured nutrient profiles of the fish in this study and all data were combined across temperature treatments ($P > 0.05$; Fig 1). However, several effects ($P < 0.05$) were observed with the brining method.

Brining resulted in a higher ash concentration with a relative increase of 28% ($P < 0.05$). Moreover, 6% loss on the overall protein concentration was noticed due to brining ($P < 0.05$). The body fat contents of tambaqui fish were not affected by the addition of salt and temperature treatments ($P > 0.05$; Table 1).

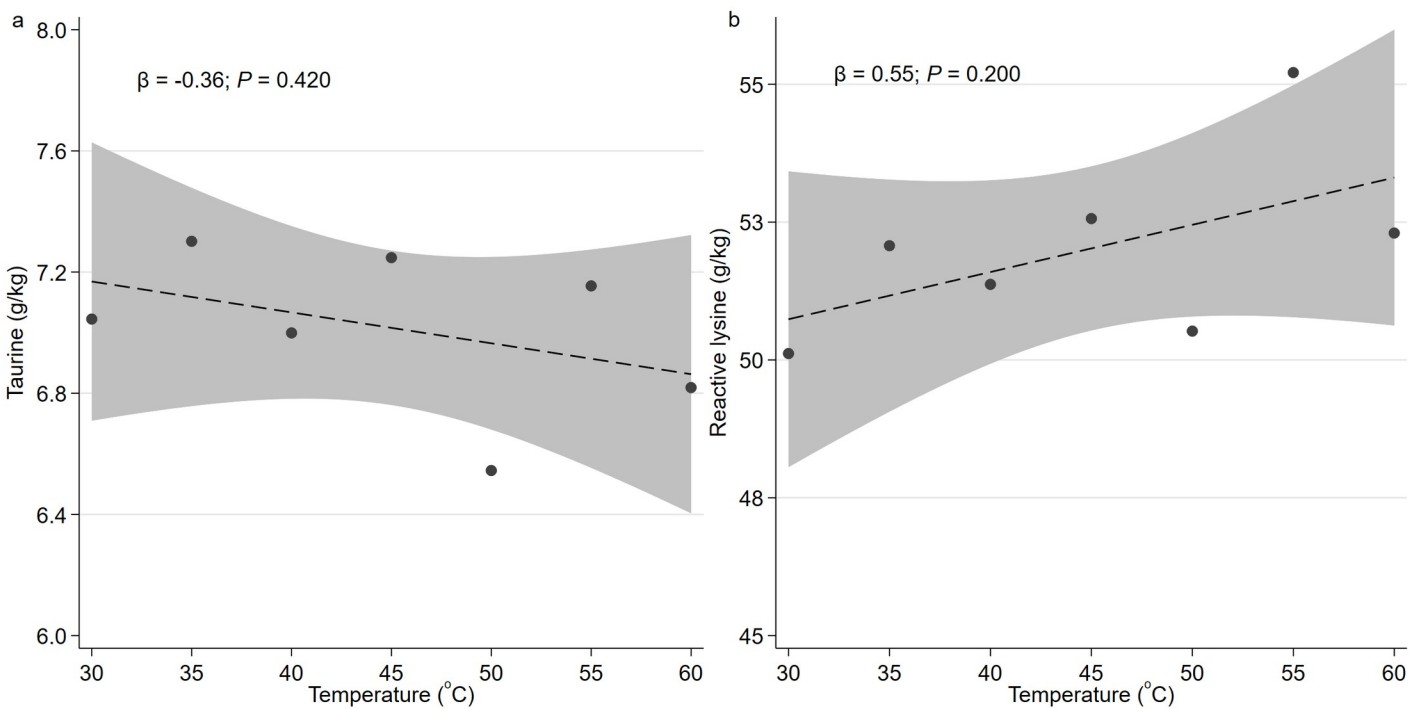

**Fig 1. The effect of drying temperature on nutrient profiles of tambaqui fish.** (a) gram of taurine per kg fish; (b) gram of reactive lysine per kg fish.

## Minerals and amino acids

The essential minerals Se and I, and the macrominerals K and Mg were significantly reduced by brining with 14, 38, 42 and 18% percent losses, respectively ($P < 0.05$). Moreover, brining resulted in an obvious enormous significant increase (744%) in Na concentration ($P < 0.05$). Otherwise, all minerals (Fe, Mn, P and Ca) were decreased by brining (Table 2).

Brining negatively affected the overall sum amino acid profile ($P < 0.05$). In particular, taurine, histidine, cysteine, methionine and available (reactive) lysine were affected by brining with the highest percent loss accounted for taurine (56%) followed by methionine (17%), cyteine (11%), and histidine (10%) ($P < 0.05$). In contrast, there was numerical 5 and 6% increase in proline and glycine respectively, whereas alanine and arginine were not affected by brining ($P > 0.05$; Table 3).

**Table 1. Changes in macronutrient concentrations in tambaqui fish (on %DM basis) due to brining and drying at seven different temperatures (30, 35, 40, 45, 50, 55 and 60°C).**

| Concentration (g/100 g DM) | Not salted | Salted | Total | Relative % Change | SEM | p | |
|---|---|---|---|---|---|---|---|
| | | | | | | salted | temp |
| **Ash** | 12.4 | 16.0 | 14.1 | 28 | 0.5 | < 0.001 | 0.793 |
| **Crude protein** | 58.8 | 55.4 | 57.0 | -6 | 0.6 | 0.004 | 0.846 |
| **Crude fat** | 28.3 | 28.7 | 28.7 | 1 | 0.4 | 0.716 | 0.657 |

DM = Dry matter; SEM = standard error of the means; temp = temperature.

Values represent mean and standard error of the means based on three replicates per treatment and seven treatments; n = 48.

Table 2. Changes in mineral concentrations in tambaqui fish due to brining and drying at seven different temperatures (30, 35, 40, 45, 50, 55 and 60˚C).

| Minerals | Not salted | Salted | Total | Relative % Change | SEM | p | |
|---|---|---|---|---|---|---|---|
| | | | | | | salted | temp |
| **Macrominerals (g/kg DM)** | | | | | | | |
| Ca | 37.4 | 37.1 | 37.0 | -1 | 1.1 | 0.965 | 0.965 |
| K | 9.71 | 5.61 | 7.71 | -42 | 0.6 | <0.001 | 0.495 |
| Mg | 1.32 | 1.10 | 1.10 | -18 | 0.03 | <0.001 | 0.880 |
| Na | 2.70 | 22.9 | 12.1 | 744 | 2.7 | <0.001 | 0.564 |
| P | 21.7 | 20.4 | 21.0 | -6 | 0.6 | 0.317 | 0.814 |
| **Microminerals (mg/kg DM)** | | | | | | | |
| Fe | 42.0 | 36.9 | 39.2 | -12 | 2.4 | 0.256 | 0.188 |
| I* | 604 | 373 | 489 | -38 | 49 | 0.020 | 0.264 |
| Mn | 9.80 | 8.81 | 9.31 | -10 | 0.3 | 0.115 | 0.152 |
| Se* | 149 | 128 | 137 | -14 | 5 | 0.020 | 0.449 |
| Zn | 48.5 | 50.6 | 49.6 | 4 | 1.3 | 0.473 | 0.588 |

DM = Dry matter; SEM = standard error of the means; temp = temperature.

Values represent mean and standard error of the means based on three replicates per treatment and seven treatments; n = 48.

*Values are presented in µg/kg DM.

## Multivariate classification of variables

Principal component analysis was computed on the collective factors obtained to appraise the relationships of nutrient profiles based on their responses towards brining and drying methods. By examining at the scree plot and eigenvalues and looking for a point at which the proportion of variance explained by each subsequent principal component drops off, an elbow in the screen plot, the most explaining two components 1 (44.8%) and 2 (18.9%) were considered and a biplot PCA generated.

In line, there were no associations between nutrient profiles and drying temperature, while several associations were observed with brining. Negative association was seen between brining, Na and ash on one side of component 1 and most other nutrients on the other. However, positive association between the brining proxies (Na, ash) and amino acids prominent in connective tissue (glycine, proline, arginine, alanine) were typically seen clustered together right of the component. The more central position of Zn, Fe, fat, and available Lys point to their low response to brining and temperature. More importantly, temperature itself was positioned centrally in the plot, emphasizing the absence of drying temperature effects on the study parameters (Fig 2).

## Discussion

The present study evaluated the effect of brine salting (15% *w/v*) prior to drying at different temperatures on the nutrient profiles of tambaqui fish. Brining induced a prominent effect on macronutrient and micronutrient profiles of tambaqui fish.

In this study, brining induced pronounced losses of specific amino acids and minerals, irrespective of drying temperature. Gill et al. [48] found that salting enhanced the aggregation of myosin in cod, yet only above a drying temperature of 50˚C. Marínez-Alvarez and Gómez-Guillén [33] indicated that there is a notable loss of soluble muscle proteins (particularly, actin and myosin) by osmosis in salting (NaCl) at pH 6.5 on cod muscle (*Gadus morhua*). Differences between studies can evolve from a wide range of conditions used in the drying process,

**Table 3. Changes in amino acid concentrations in tambaqui fish due to brining and drying at seven different temperatures (30, 35, 40, 45, 50, 55 and 60°C).**

| Amino acids (g/kg CP) | Not salted | salted | Total | Relative % Change | SEM | p | |
|---|---|---|---|---|---|---|---|
| | | | | | | salted | temp |
| Taurine | 7.10 | 3.10 | 5.11 | -56 | 0.5 | <0.001 | 0.324 |
| Asparagine + Aspartic acid | 56.8 | 54.9 | 55.4 | -3 | 0.9 | 0.268 | 0.671 |
| Threonine | 27.0 | 26.3 | 26.5 | -3 | 0.4 | 0.311 | 0.965 |
| Serine | 25.0 | 24.9 | 24.8 | 0 | 0.2 | 0.807 | 0.777 |
| Glutamine + Glutamic acid | 91.4 | 88.3 | 89.2 | -3 | 1.3 | 0.247 | 0.851 |
| Glycine | 75.7 | 80.2 | 79.3 | 6 | 2.5 | 0.361 | 0.750 |
| Alanine | 51.1 | 51.0 | 51.2 | 0 | 0.7 | 0.961 | 0.636 |
| Valine | 30.8 | 29.2 | 29.8 | -5 | 0.5 | 0.062 | 0.642 |
| Isoleucine | 26.0 | 24.6 | 24.9 | -5 | 0.5 | 0.162 | 0.838 |
| Leucine | 44.3 | 41.8 | 42.5 | -6 | 0.9 | 0.134 | 0.855 |
| Tyrosine | 17.9 | 16.5 | 16.9 | -8 | 0.5 | 0.098 | 0.952 |
| Phenylalanine | 24.5 | 23.4 | 23.8 | -5 | 0.4 | 0.134 | 0.874 |
| Histidine | 15.5 | 14.0 | 14.6 | -10 | 0.4 | 0.054 | 0.400 |
| Lysine | 51.1 | 47.4 | 48.6 | -7 | 1.1 | 0.070 | 0.863 |
| Arginine | 48.3 | 48.3 | 48.5 | 0 | 0.5 | 0.999 | 0.464 |
| Proline | 47.5 | 49.9 | 49.5 | 5 | 1.3 | 0.356 | 0.660 |
| Cysteine | 5.10 | 4.40 | 4.60 | -11 | 0.1 | 0.016 | 0.746 |
| Methionine | 14.7 | 12.1 | 13.3 | -17 | 0.4 | <0.001 | 0.675 |
| Tryptophan | 5.10 | 4.60 | 4.71 | -8 | 0.2 | 0.229 | 0.698 |
| Reactive lysine | 52.0 | 46.4 | 48.5 | -11 | 1.2 | 0.004 | 0.851 |
| % Reactive lysine | 102.2 | 97.9 | 99.9 | -4 | 1.8 | 0.306 | 0.760 |
| Sum AA | 665 | 645 | 653 | -3 | 6.87 | 0.017 | 0.621 |

CP = Crude protein; AA = Amino acid; SEM = Standard error of the means; temp = temperature.

Values represent mean and standard error of the means based on three replicates per treatment and seven treatments; n = 48.

[a]Percentage of reactive lysine in the total protein.

[b]Percentage of reactive lysine in the total lysine.

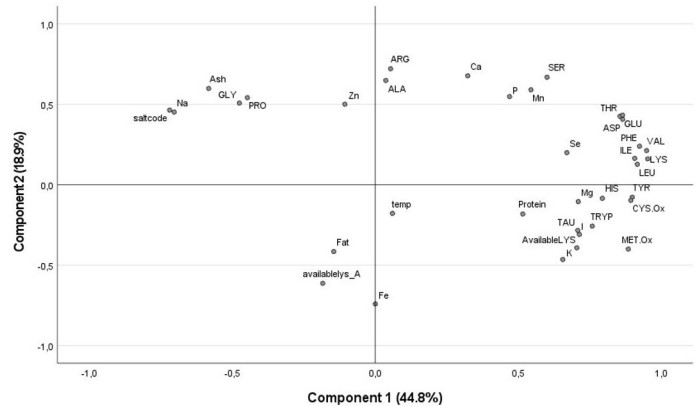

**Fig 2. Principal component analysis classification of nutrient profiles based on their response to brining and temperature.** Analyses represent principal component analysis (PCA) of evaluated nutrient values (expressed as percentages of the sum of total) after brining and drying of whole tambaqui fish samples (n = 48). AvailableLYS, is the percentage of reactive lysine in the total protein; availablelys_A, is the percentage of reactive lysine in the total lysine; saltcode, refers to brining; temp, is the drying temperature.

including the drying type, drying temperature and duration, relative humidity, air velocity and the specific characteristics of the product dried (species and size of the fish). Although comparison with existing literature does not allow us to pinpoint the factors that protected from a drying temperature effect, this study suggests that brining is associated with nutrient losses. In the present study, a total of 6% crude protein was lost due to the brining stress. In fact, salt concentration and salting duration can modulate the effect of brining on muscular protein loss in fish [28–30]. The solubility of muscle proteins often start to decrease with increasing salt concentration due to protein aggregation [24] and/or altered enzymatic activities [49]. In the present study, however, an acceptable (15% *w/v*) salt was used for brining [31].

The amino acid analysis in this study demonstrated that the salt-induced protein loss was not an overall effect, but reduced specific amino acids, taurine, histidine, cysteine, methionine and available lysine. Taurine is not involved in protein synthesis but often accumulates in muscle tissues of fish and has osmolytic properties [50, 51]. Hence, the loss in taurine seem a logical reaction to the osmotic stress from brining. This taurine loss can be important since its role in health has been demonstrated for instance in infants [52] and it is also the main amino acid to form conjugated bile acids in chickens [53].

The tendency to a lower histidine concentration due to brining can be seen in the same light. Apart from its protein-bound appearance, histidine is stored in muscular tissue as either free histidine (acetylated or not) or bound to mainly beta-alanine (hence constituting dipeptides, such as carnosine or anserine); this storage of histidine occurs throughout the animal kingdom [54]. Shiau [55] reported that histidine can easily get released from muscle tissues during stress conditions, such as starvation. Significant losses of the free amino acids histidine, taurine and glycine were reported in salted milkfish [32]. Histidine is involved in the regulation and metabolism of essential trace minerals, formation of enzymes, induction of the immune system through the production of histamine and protects brain cells through the formation of metallothionein among others [56].

Cysteine and methionine are sulphur-containing amino acids and quite reactive to processing, in that they could either get directly affected by the ionic brine solution or can possibly make cross-links that reduce their tissue availability [32]. Kumazawa [57] reported that brining may cause cross-linking reactions of muscle myosin heavy chains and form ε-(γ-glutamyl) lysine through transglutaminase. There was no effect on the percentage of reactive lysine (% available-lysine), meaning that both salting and drying did not induce Maillard reactions that affect lysine availability. The relative decrease in the total lysine likely explains the decreased percentage of available lysine in the current study.

Four specific amino acids in the principal component analysis separated from the negative association between brining and the other amino acids: alanine and arginine were not affected, while glycine and proline were even positively associated with brining. These amino acids are prominent in the skeleton and in scales, which is a strong indication that the highest protein losses were from muscle and other soft tissues [58]. Other authors not always report similar changes in amino acid pattern: a sigmoid initial increase (up to 9 h) and final slight decrease (48 h) of alanine, glutamine and lysine was reported in salted milkfish [32]; alanine, glycine, lysine and histidine increased, when salted fish was dried [59]. The increase in these amino acid concentrations could be due to the decrease in the proportion of the extracted total lipids present as polyunsaturated fatty acids as depicted by Yankah [59]. Fat is a hydrophobic molecule and can act as a physical barrier to limit salt (NaCl) and water transport across the muscle tissue [23]. In fact, the lipid content was higher for the fish in our study (28.71%) although fat concentration was not affected by both treatments. Of the affected amino acids by brining, taurine and lysine are polar hydrophilic where are methionine and cysteine have basic hydrophobic property [47, 50, 51].

The specific loss of soft tissue from the fish is not only supported by the amino acid data, but also witnessed by the change in mineral profile. The considerable loss of the microminerals Se and I and the macrominerals K and Mg following brining reflects the presence of these minerals in soft tissue [60]. Selenium typically occurs as replacement of sulphur in sulphur-containing amino acids (cysteine and methionine) in soft tissues. The decrease in methionine and cysteine seen in our study, therefore, fits the observation of decreased selenium. In contrast, the typical skeletal minerals, such as Ca, P and Mg, and those that are deposited in scales and fins such as Zn and Fe, were not affected. Potassium is a strong electrolyte that could disappear in the brine solution or get filtered out in exchange with the added sodium. The loss of water due to salt was explained as the main cause for leaching of these macrominerals in the solution during brining. The decrease in K concentration agrees with Polak-Juszczak [61] and Bakhiet [62] from salted fish. The latter study remarkably reported a salt-induced increase in iodine concentration, most likely evolving from salt impurities as proposed by the authors. It is not clear how these differences with our study can be explained, but we used a purified source of salt, hence excluding potential unintended supply of microminerals to the fish tissues through brining.

Drying is an energy intensive process, which can use about 10% or more of the total energy consumed during processing. In the present study, although our aim was not to quantify the drying efficiency, we conjointly were aiming to consume less energy while drying samples. This was achieved by increasing the sample surface area of exposure, avoiding surface contact by putting beads, improved equipment insulation during and between drying, supply of a constant air ventilation at 1.0 m/s during drying, and minimizing the timing between dryings [63].

Undeniably, brining has substantially increased Na and ash concentrations. This has been observed in most studies on brining of fish [61, 62]. This enrichment with salt through brining must be considered when formulating diets but, more importantly, our results demonstrate that brining may not be the best choice to maintain the nutritive value of fish. Evidently, we must acknowledge that nutritive value is not the only criterion to pick a preservation method for fish: for the reduction of the pathogenic load in fish products, salt addition may still be a valuable technique that goes, however, at the expense of nutritive value as shown in our study. A comforting finding is that the wide range in drying temperature evaluated in our study had no noticeable impact on the nutritive value of the fish.

## Conclusions

This research shows that drying temperature has no effect on nutrient composition of the studied fish which could be encouraging for drying of fish within the drying temperature range used in this study. Brining, however, reduced some important trace elements such as Se and I, and lead to loss of muscle crude protein, with the concomitant amino acids. Brining also increased the leaching of K, I, and taurine. The formation of Maillard products was not promoted by both drying and brining treatments, as demonstrated through unaltered reactive lysine proportions. Therefore, drying without brining may better preserve the nutritive value of tambaqui fish. However, as a recommendation to the industry sector, it is suggested that the final products may further evaluated for any pathogen of economic or public health importance.

## Supporting information

**S1 Table. Effect of individual drying temperatures on nutrient profiles of tambaqui fish.** (DOCX)

**S1 File. Excel sheet raw datasets for brining and drying temperatures.**
(XLSX)

**S2 File. Data oupt1 for brining and drying temperatures on SPSS.**
(SPV)

**S3 File. Data oupt2 and graphs for brining and drying temperatures on SPSS.**
(SPV)

## Acknowledgments

The authors acknowledge Herman De Rycke for the proximate analysis.

## Author Contributions

**Conceptualization:** Awot Teklu Mebratu, Yohannes Tekle Asfaw, Wouter Merckx, Wouter H. Hendriks, Geert P. J. Janssens.

**Data curation:** Awot Teklu Mebratu, Wouter Merckx, Geert P. J. Janssens.

**Formal analysis:** Awot Teklu Mebratu, Geert P. J. Janssens.

**Funding acquisition:** Geert P. J. Janssens.

**Investigation:** Awot Teklu Mebratu.

**Methodology:** Awot Teklu Mebratu, Yohannes Tekle Asfaw, Wouter Merckx, Geert P. J. Janssens.

**Project administration:** Wouter Merckx, Geert P. J. Janssens.

**Resources:** Wouter Merckx, Wouter H. Hendriks.

**Supervision:** Yohannes Tekle Asfaw, Geert P. J. Janssens.

**Validation:** Yohannes Tekle Asfaw, Wouter Merckx, Wouter H. Hendriks, Geert P. J. Janssens.

**Writing – original draft:** Awot Teklu Mebratu.

**Writing – review & editing:** Awot Teklu Mebratu, Yohannes Tekle Asfaw, Wouter Merckx, Wouter H. Hendriks, Geert P. J. Janssens.

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
