## [Decision Letter · Decision Letter 0]

31 Aug 2023

PONE-D-22-35077A laboratory simulation of the impact of brining and drying as accessible preservation strategies under tropical rural conditions on the nutritive value of tambaqui fish (Colossoma macropomum)PLOS ONE

Dear Dr. MEBRATU,

Thank you for submitting your manuscript to PLOS ONE. After careful consideration, we feel that it has merit but does not fully meet PLOS ONE’s publication criteria as it currently stands. Therefore, we invite you to submit a revised version of the manuscript that addresses the points raised during the review process.

ACADEMIC EDITOR:The manuscript has been reviewed and requires a major revision; all Editor's and reviewers' comments should be carefully considered in order to improve its quality.

We look forward to receiving your revised manuscript.

Kind regards,

Emmanuel Oladeji Alamu

Academic Editor

PLOS ONE

Additional Editor Comments:

The manuscript has been reviewed and requires a major revision; all Editor's and reviewers' comments should be carefully considered in order to improve its quality. Please check the uploaded files for the comments.

Reviewers' comments:

Reviewer's Responses to Questions

**Comments to the Author**

1. Is the manuscript technically sound, and do the data support the conclusions?

Reviewer #1: Yes

Reviewer #2: Yes

Reviewer #3: Yes

2. Has the statistical analysis been performed appropriately and rigorously? 

Reviewer #1: No

Reviewer #2: Yes

Reviewer #3: Yes

3. Have the authors made all data underlying the findings in their manuscript fully available?

Reviewer #1: No

Reviewer #2: Yes

Reviewer #3: Yes

4. Is the manuscript presented in an intelligible fashion and written in standard English?

Reviewer #1: Yes

Reviewer #2: Yes

Reviewer #3: Yes

5. Review Comments to the Author

Reviewer #1: Manucript title: A laboratory simulation of the impact of brining and drying as accessible preservation strategies under tropical rural conditions on the nutritive value of tambaqui fish

(Colossoma macropomum)

GENERAL COMMENTS

The paper is well written with strong justification and very good elaboration in methods. However some of the sections such as data analysis is not well elaborated. The types of statistics used needs to be appropriate to answers the objective of the research.

OTHER COMMENTS

Abstract

In the abstract introduce the concept of brine salting is what, before “The present study evaluated…..”

Introduction

Line 93: You could add “macronutrients and micronutrients”

Materials and Methods

Line 97-104: The paragraph needs to be cited.

Line 120: A total of 48 tambaqui fish (792 ± 16 g; 8 months old). This number os not clear, whether say it mean weight (sd) or provide unit explanation.

Statistical analysis

Line 213-215: Univariate analysis is done for each separate variable in a descriptive way. Here did you mean bivariate analysis for comparing variance? Please clarify

Line 215: “All data were presented as means of three values and standard error of the means (SEM)”. Is not clear, What does means of three value imply for?

Where are the results of Principal Component Analysis (PCA)? Why did the researcher opt to use PCA in this research?

Results

You could better start the results section by provide a small descriptive analysis of the samples and general information about your data.

Line 227: the tile of Fig 1 is there, but figure is not shown. Place the figure header below the figure.

Table 1: The table generalizes the values of macronutrients at different temp, could you try and present at each temp and we could see the variations? Table 1 needs to be improved, I can only see the values during salting, but not drying temp.

Label below the table have to be clearly presented such as DM=Dry matter.

Line 239: The essential “microminerals”. Just say micronutrients or minerals

Table 2: Similar to table 1, different temp values needs to be presented in table

Line 264: Multivariate statistical analysis (PCA), revise either write PCA on remain with multivariate

Multivariate classification of variables: This section is not well written and confusing. Which association did the author is presenting? PCA is not showing the association between variables. If multivariate analysis was used which type of statistical technique was applied. Just to say the association is not enough, it has to be presented with values. What makes you say this is association/no association? Please clarify.

Discussion

Start by providing the objective the research and summarizing the main findings.

Line 283: In our study. Use “In this study”

How these literature compared here related to Tambaqui fish? Or related to included sample? Compare and contrast with relevant papers/research.

Line 292: more than 60C. You don’t have that temp in your study. The final temp is 60 Centigrade.

Line 298: it is clear that brining completely overruled the drying temperature in its effect on nutrient loss. Just use the term such as “ This study suggests that brining is associated with nutrients loss…..”.

Line 299: 6% of tissue protein. Focus on what you measure “Crude protein”

The limitation and strenght of this research is very important to be stated in this research.

Conlcusion

Conclusion should be straigh and clear. Please consider revising. Example, this research shows that ……

Reviewer #2: Line 146. Please specify the model of oven used and Humidity condition set for laboratory simulation.

Line 153. The statement (Samples were finally freeze-dried) looks contradictory with the title "A laboratory simulation of the impact of brining and drying as accessible preservation strategies under tropical rural conditions on the nutritive value of tambaqui fish (Colossoma macropomum)"

Line 376. The language of the conclusion must be improved for better understanding of the reader. This section must be sound.

Reviewer #3: This is a good work and needs attention of the researchers. I must recommend it to be published.

Comments to editor

Hi Editor Journal

In the current manuscript the authors have studied "A laboratory simulation of the impact of brining and drying as accessible preservation strategies under tropical rural conditions on the nutritive value of tambaqui fish". This is a good research work and need of the hour. I recommend this work to be published in your esteemed journal.

Thanks

6. PLOS authors have the option to publish the peer review history of their article (what does this mean?). If published, this will include your full peer review and any attached files.

Reviewer #1: **Yes: **Dr. Ahmed Gharib Khamis

Reviewer #2: No

Reviewer #3: No

---

## [Author Response · Author response to Decision Letter 0]

13 Dec 2023

Dear Editor and Reviewers, 

We, the authors, are thankful for your time in reviewing our manuscript. The comments given by the reviewers and editors were highly valuable for the quality of our manuscript. Now, to the best of our knowledge, we have tried to incorporate all the comments given and address them with rebutal letter one-byone.

Thank you for your time,

---

## [Decision Letter · Decision Letter 1]

29 Jan 2024

PONE-D-22-35077R1Impact of brining and drying processes on the nutritive value of tambaqui fish (Colossoma macropomum)PLOS ONE

Dear Dr. Mebratu,

Thank you for submitting your manuscript to PLOS ONE. After careful consideration, we feel that it has merit but does not fully meet PLOS ONE’s publication criteria as it currently stands. Therefore, we invite you to submit a revised version of the manuscript that addresses the points raised during the review process.

The paper in its current state is unsuitable for publication; it requires minor revisions in accordance with the reviewers' suggestions. Please carefully attend to the reviewers' comments and ensure you do thorough grammar checks, as there are a few errors.

Please submit your revised manuscript by Mar 14 2024 11:59PM. If you will need more time than this to complete your revisions, please reply to this message or contact the journal office at plosone@plos.org. Please include the following items when submitting your revised manuscript:A rebuttal letter that responds to each point raised by the academic editor and reviewer(s). You should upload this letter as a separate file labeled 'Response to Reviewers'.A marked-up copy of your manuscript that highlights changes made to the original version. You should upload this as a separate file labeled 'Revised Manuscript with Track Changes'.An unmarked version of your revised paper without tracked changes. You should upload this as a separate file labeled 'Manuscript'.If applicable, we recommend that you deposit your laboratory protocols in protocols.io to enhance the reproducibility of your results. Protocols.io assigns your protocol its own identifier (DOI) so that it can be cited independently in the future. For instructions see: https://journals.plos.org/plosone/s/submission-guidelines#loc-laboratory-protocols. Additionally, PLOS ONE offers an option for publishing peer-reviewed Lab Protocol articles, which describe protocols hosted on protocols.io. Read more information on sharing protocols at https://plos.org/protocols?utm_medium=editorial-email&utm_source=authorletters&utm_campaign=protocols.

We look forward to receiving your revised manuscript.

Kind regards,

Emmanuel Oladeji Alamu

Academic Editor

PLOS ONE

Journal Requirements:

Additional Editor Comments:

The paper in its current state is unsuitable for publication; it requires minor revisions in accordance with the reviewers' suggestions. Please carefully attend to the reviewers' comments and ensure you do thorough grammar checks, as there are a few errors.

Reviewers' comments:

Reviewer's Responses to Questions

**Comments to the Author**

1. If the authors have adequately addressed your comments raised in a previous round of review and you feel that this manuscript is now acceptable for publication, you may indicate that here to bypass the “Comments to the Author” section, enter your conflict of interest statement in the “Confidential to Editor” section, and submit your "Accept" recommendation.

Reviewer #2: All comments have been addressed

Reviewer #4: (No Response)

2. Is the manuscript technically sound, and do the data support the conclusions?

Reviewer #2: Yes

Reviewer #4: Yes

3. Has the statistical analysis been performed appropriately and rigorously? 

Reviewer #2: Yes

Reviewer #4: Yes

4. Have the authors made all data underlying the findings in their manuscript fully available?

Reviewer #2: Yes

Reviewer #4: Yes

5. Is the manuscript presented in an intelligible fashion and written in standard English?

Reviewer #2: No

Reviewer #4: Yes

6. Review Comments to the Author

Reviewer #2: The authors contributed well in their respective field and addressed a social problem but some modifications may be needed, i.e.,

1. The abstract lacks clarity and brevity and has ambiguity in presenting the problem statement and results.

2. The introduction section is too long, with a lot of vague sentences, and is poor in its construction.

3. The material and method are against the problem statement, i.e., the use of freeze drying.

4. The result and discussion are well presented and have clarity.

5. The conclusion is improved compared to previous one and addresses the problem statement.

Reviewer #4: The authors investigated “Impact of brining and drying processes on the nutritive value of tambaqui fish (Colossoma macropomum)”. Some important findings were reported. However, the manuscript would require a minor revision to improve on its present quality. Therefore, I have annotated some of the revisions, including some missing commas, in the attached manuscript file, for an easy reference. In addition, authors are requested to kindly take note of the following suggestions.

Line 186: Please change "try" to "tray".

Lines 197-201: This highlighted sentence in these lines is too long. Please separate it into two thus "Samples were then kept at -21°C overnight and finally freeze-dried (CoolSafe™, SCANVAC, Denmark) for a period of 16 h each to a stable weight, uniformly ground (2.0 mm sieve). Subsequently, the samples were homogenized using a grinding machine (DeˊLonghi™, KG48) at 12× speed and then individually kept in closable plastic containers at 4°C until analyzed [22]."

Line 235: Please use a past tense for "are often" and specify the number of times the visual inspection was performed.

Line 236: Please change "spread chamber and mobilizer" to "spray chamber and nebulizer".

Line 237: Please change "Blanc" to "Blank".

Line 242: Please change "Sulphur containing" to "Sulphur-containing".

Line 278: Please change "Proximate analysis" to "Proximate composition".

Line 379: Please change "reduces" to "reduce".

Lines 394-396: Sequel to the highlighted statement in lines 394-396, the authors can further explain which class of amino acids (whether hydrophobic, polar, acidic, basic, neutral) was most affected by brining.

Lines 415-419: The sentence in lines 415-419 is too long. Please split it into two.

Lines 428-430: The authors of this study, used controlled temperatures. So, how can they claim that "This implies that drying of fish under uncontrolled environmental temperature seems a suitable option in rural areas to preserve the nutritive value of (tambaqui) fish"? If Maillard reaction products are formed between the amino acid residues and reducing sugars in the fish due to a high "uncontrolled environmental temperature", can this claim (that is, the preservation of the nutritive value of the fish) still hold?

Please note that Maillard reaction products' formation can increase with increasing drying temperature. You may refer to https://doi.org/10.1016/j.fshw.2019.03.012

Lines 432-435: Sequel to the preceding line of reasoning, please be cautious in the first sentence (lines 432-435) in the conclusion. I suggest adding "within the drying temperature range used in this study".

7. PLOS authors have the option to publish the peer review history of their article (what does this mean?). If published, this will include your full peer review and any attached files.

Reviewer #2: **Yes: **Muhammad Amir

Reviewer #4: **Yes: **Emmanuel Anyachukwu Irondi

---

## [Author Response · Author response to Decision Letter 1]

2 Feb 2024

Dear Reviewers, 

The round-2 comments given by the reviewers were highly valuable for the quality and clarity of our manuscript. Now, to the best of our knowledge, we have tried to incorporate all the comments given so far including gramatical and typographical corrections and we have responded to them point-by-point in a separate rebuttal letter annexed as 'Response to Reviewers'.

We really appreciate!

---

## [Decision Letter · Decision Letter 2]

13 Feb 2024

PONE-D-22-35077R2Impact of brining and drying processes on the nutritive value of tambaqui fish (Colossoma macropomum)PLOS ONE

Dear Dr.Mebratu,

Thank you for submitting your manuscript to PLOS ONE. After careful consideration, we feel that it has merit but does not fully meet PLOS ONE’s publication criteria as it currently stands. Therefore, we invite you to submit a revised version of the manuscript that addresses the points raised during the review process.

Thank you for the good work, and the paper is looking good now. However, as Reviewer 2 pointed out in the attached reviewed version, there are a few minor corrections that are necessary. Please attend to these corrections as quickly as possible.

We look forward to receiving your revised manuscript.

Kind regards,

Emmanuel Oladeji Alamu

Academic Editor

PLOS ONE

Journal Requirements:

Additional Editor Comments:

Thank you for the good work, and the paper is looking good now. However, as Reviewer 2 pointed out, there are a few minor corrections that are necessary. Please attend to these corrections as quickly as possible.

Reviewers' comments:

Reviewer's Responses to Questions

**Comments to the Author**

1. If the authors have adequately addressed your comments raised in a previous round of review and you feel that this manuscript is now acceptable for publication, you may indicate that here to bypass the “Comments to the Author” section, enter your conflict of interest statement in the “Confidential to Editor” section, and submit your "Accept" recommendation.

Reviewer #2: All comments have been addressed

Reviewer #4: All comments have been addressed

2. Is the manuscript technically sound, and do the data support the conclusions?

Reviewer #2: Yes

Reviewer #4: Yes

3. Has the statistical analysis been performed appropriately and rigorously? 

Reviewer #2: Yes

Reviewer #4: Yes

4. Have the authors made all data underlying the findings in their manuscript fully available?

Reviewer #2: Yes

Reviewer #4: Yes

5. Is the manuscript presented in an intelligible fashion and written in standard English?

Reviewer #2: Yes

Reviewer #4: Yes

6. Review Comments to the Author

Reviewer #2: The author addressed all the comments, and the data presented is technically sound, have no conflict with ethical policy.

Reviewer #4: The authors have satisfactorily revised the paper in line with the recommendation made earlier, improving it quality.

7. PLOS authors have the option to publish the peer review history of their article (what does this mean?). If published, this will include your full peer review and any attached files.

Reviewer #2: **Yes: **Muhammad Amir

Reviewer #4: **Yes: **Emmanuel Anyachukwu Irondi

---

## [Author Response · Author response to Decision Letter 2]

15 Feb 2024

Dear Reviewers, 

The all round comments given by the reviewers were highly valuable for the further quality of our manuscript. Now, to the best of our knowledge, we have tried to incorporate all the minor comments and corrections given by Revier-2 so far and responded to them point-by-point in a separate rebuttal letter, "Response to Reviewers".

We really appreciate for your time and scientific contribution.

On behalf of the Authors,

---

## [Editor Report · Decision Letter 3]

19 Feb 2024

Impact of brining and drying processes on the nutritive value of tambaqui fish (Colossoma macropomum)

PONE-D-22-35077R3

Dear Dr. Mebratu,

We’re pleased to inform you that your manuscript has been judged scientifically suitable for publication and will be formally accepted for publication once it meets all outstanding technical requirements.

Kind regards,

Emmanuel Oladeji Alamu

Academic Editor

PLOS ONE
---

## [Editor Report · Acceptance letter]

28 Mar 2024

PONE-D-22-35077R3 

PLOS ONE

Dear Dr. MEBRATU, 

I'm pleased to inform you that your manuscript has been deemed suitable for publication in PLOS ONE. Congratulations! Your manuscript is now being handed over to our production team.

Kind regards, 

on behalf of

Dr. Emmanuel Oladeji Alamu 

Academic Editor

PLOS ONE